# Canadian status of "not acceptable" drugs as evaluated by Prescrire: A cohort study

Joel Lexchin[1,2]*

1 School of Health Policy and Management, York University, Toronto, Canada, 2 Department of Family and Community Medicine, University of Toronto, Toronto, Canada

* jlexchin@yorku.ca

## Abstract

### Introduction

The independent French drug bulletin Prescrire International rates the therapeutic innovation of new drug-indications approved for marketing in France using an ordinal scale with the lowest rating being "not acceptable". This study investigates whether these drugs were approved by Health Canada.

### Methods

A list of "not acceptable" drug-indications was generated by handsearching all issues of Prescrire International between January 2013 and December 2022. The generic names, indications and reasons why Prescrire labeled them not acceptable were recorded. The approval date was determined by consulting the website of the European Medicines Agency (EMA). The status of these drug-indications in Canada was determined by searching multiple Health Canada websites. Therapeutic Evaluations for new drug-indications done by the Patented Medicine Prices Review Board (PMPRB) were recorded.

### Results

Prescrire rated 57 new drug-indications and 42 new indications for existing drugs as not acceptable. Seventy of these drug-indications were available in Canada– 42 new drug-indications and 28 new indications for existing drugs. Twenty (90.9%) of the 22 new drugs evaluated by the PMPRB were rated as slight/no therapeutic improvement and 2 as moderate therapeutic improvement. The median difference, in days, between approval times by the EMA/ANSM and Health Canada was 129 (interquartile range -102, 341) in favour of the former.

### Discussion

The majority of the not acceptable drug-indications were approved by Health Canada. The difference between when Prescrire and Health Canada examined the evidence for these drug-indications is unlikely to explain the difference in their evaluations. A change in regulatory standards at Health Canada may be one factor behind the presence of these drugs. To what degree those drugs led to more harms than benefits for patients who are taking them

**Data Availability Statement:** All data submitted as Supporting information file.

**Funding:** The author(s) received no specific funding for this work.

**Competing interests:** Between 2020-2024, Joel Lexchin received payments for writing a brief on the role of promotion in generating prescriptions for a legal firm, for being on a panel about pharmacare and for co-writing an article for a peer-reviewed medical journal. He is a member of the Board of Canadian Doctors for Medicare. He receives royalties from University of Toronto Press and James Lorimer & Co. Ltd. for books he has written. He is participating in research funded by the Canadian Institutes of Health Research. This does not alter our adherence to PLOS ONE policies on sharing data and materials.

needs to be urgently investigated. Finally, the reasoning behind Health Canada's approval of these drugs should be interrogated.

## Introduction

The independent French drug journal Prescrire International, a monthly medical journal that reviews new treatments for healthcare professionals in France and internationally [1], rates the therapeutic innovation of new drug-indications (new drugs and new indications for existing drugs) approved for marketing in France both by the centralized European Medicines Agency (EMA) procedure and by the EMA's decentralized procedure which allows a company to apply for approval in a single market. A drug-indication is defined as a specific use for a particular drug, for example, the use of drug X for the treatment of metastatic non-small cell lung cancer.

Prescrire assesses therapeutic value through a multistep process. First, it "examines the condition or clinical setting for which the drug is proposed; then the natural course of the disease, the efficacy and safety of existing treatments, and the most relevant outcome measures. This is followed by a systematic search for clinical data on the efficacy and adverse effects of the new drug, and an assessment of the level of evidence. Based on [its] independent analysis of clinical data, [it] form[s] a judgement as to whether the new drug is beneficial for patients or whether or not its harmful effects outweigh the benefit." Based on its analysis, it rates products using a six-point ordinal scale ranging from "bravo", a major therapeutic innovation in an area where previously no treatment was available to "not acceptable", that is, without evident benefit but with potential or real disadvantages. A seventh category is "judgment reserved" where the decision is postponed until better data and a more thorough evaluation can be made [2]. Although bodies such as the French regulatory agency Haute Autorité de Santé (HAS) and the Institute for Quality and Efficiency in Health Care, the German health technology assessment agency, also evaluate the therapeutic value of new drugs using an ordinal scale and publish their findings in English, the evaluations from Prescrire are used in this study because it is the only organization that names drugs that should not have been given regulatory approval. Prescrire's ratings generally are in line with those from other organizations such as the bulletin of the Swedish Medical Products Agency [3] and the HAS [4, 5].

According to Prescrire, not acceptable drug-indications should never have been marketed because of their negative benefit to harm ratio. Their presence on a national market could be seen as a measure of the rigour of the drug approval process in the jurisdiction and as an indication of possible harms to patients using them. Previous work has investigated the Canadian status of drugs that Prescrire recommended should be avoided [6]. The purpose of this study is to examine whether not acceptable drug-indications were approved by Health Canada.

## Methods

### List of not acceptable drugs

The 110 issues of Prescrire International published between January 2013 and December 2022 were hand searched between March 12–16, 2024 by a single individual to construct a list of drug-indications that were labeled as not acceptable. For each drug, the generic name, brand name, indication, whether it was a new drug versus a new indication, the formulation of the drug and the reason(s) that Prescrire considered it should not be marketed were extracted and the data were entered into an Excel spreadsheet. (The formulation of a drug may affect its ease

of use, how quickly it is absorbed and how often it needs to be used. Differences in these characteristics may contribute to different assessments of whether a drug should be on the market.) The date of approval of the drug-indications marketed in France was found by searching on the drug's name on either the EMA website [7] or the website of the French Agence nationale de séurité du medicament et des produits de santé (ANSM) [8].

## Canadian data

In order to determine if the drug-indication was available on the Canadian market its generic name was entered into the search function on the Drug Product Database [9]. If the drug was found, then the Product Monograph, accessible from the Drug Product Database, was searched to see if one of the indications matched the one assessed by Prescrire. If it was a new drug, then the Notice of Compliance website was consulted for the approval date. If it was a new indication for an existing drug then the Summary Basis of Decision website which describes the postmarket regulatory activity for drugs approved since June 2012 was used to find the date and type of approval of the new indication [10]. Since December 2015 and January 2017 Health Canada has been listing the names of new drugs and new indications, respectively, that were not approved on a website and that website was searched for any not acceptable drugs in those categories [11]. Finally, the website listing new drugs and new indications under review was searched to see if approval applications were submitted for drugs not on the Database [11].

An assessment was made of the additional therapeutic value and the safety of not acceptable drugs on the Canadian market. Accordingly, the Recall and Safety Alerts website was searched for safety advisories issued by Health Canada for drugs in this group [12] accompanied by a search of therapeutic evaluations done by the Patented Medicine Prices Review Board (PMPRB) and published in its annual reports [13]. The PMPRB is an independent quasi-judicial body that reviews the prices of new patented medicines sold in Canada and as part of that function undertakes therapeutic evaluations and categorizes the drugs as breakthrough, substantial improvement, moderate improvement, slight or no improvement. (The PMPRB does not evaluate the therapeutic value of new indications for existing drugs).

All Canadian data were entered into the same Excel spreadsheet as the data from Prescrire.

## Therapeutic classifications

Drugs were put in the second level anatomic-therapeutic-chemical category based on the World Health Organization system [14].

## Data analysis

The number of not acceptable drug-indications approved by Health Canada is reported. Counts were made of the number of drugs that had therapeutic evaluations by the PMPRB and that had safety warnings issued. Prescrire evaluations and Health Canada approvals may have differed due to the two organizations being presented with different information. Prescrire evaluations are undertaken shortly after EMA approval and therefore, the difference between the EMA and Health Canada approval dates (in days) was calculated and the median and interquartile range was reported to explore this possibility.

## Ethics and patient involvement

All data were publicly available and ethics approval was not required. No patients were involved in this study.

## Results

### Prescrire rating of not acceptable drug-indications

Over the 10-year period, January 2013 to December 2022, Prescrire rated 57 new drug-indications and 42 new indications for existing drugs as not acceptable (85 unique drugs, 99 drug-indications); 3 drugs had 3 indications and 8 drugs had 2 indications. Table 1 gives the generic names and indications of these 99 drug-indications. The plurality of the drug-indications was in 3 anatomic-therapeutic-chemical groups: antineoplastic agents = 29, drugs used in diabetes = 10, immunosuppressants = 10. In all cases, Prescrire deemed the drug-indications not acceptable because of a negative benefit to harm ratio. Eighty drug-indications had little to no efficacy but considerable harms and the remaining 19 were rated not acceptable primarily because of the harms associated with the drug-indication. Ninety-four drug-indications were approved by the EMA and 5 received French authorization through the national procedure. The approval date could not be found for 2 drug-indications. Two drugs approved by the EMA were subsequently withdrawn from the market because of safety concerns. (S1 Table gives the complete data collected for this study).

### Not acceptable drugs approved by Health Canada

Out of the 99 drug-indications identified by Prescrire as not acceptable, a total of 70 were approved by Health Canada and analyzed– 28 new indications for existing drugs (for 1 drug only 1 of 2 indications evaluated by Prescrire was approved by Health Canada) and 42 new drug-indications for a total of 62 unique drugs. Out of the remaining 29 drug-indications, 13 were not found on the Drug Product Database for one of three reasons: 9 were never approved by Health Canada, 3 applications were withdrawn by the company before Health Canada reached a decision and 1 drug had been removed from the market for safety reasons and was never submitted to Health Canada for approval. (One of the drugs not on the Database was under review by Health Canada but the indication was not given). The remaining 16 drugs were not approved for either the same indication or in the same formulation as the one evaluated by Prescrire (Fig 1).

### Canadian evaluations of the therapeutic benefits and harms of not acceptable drug-indications

Twenty-two new drugs were evaluated by the PMPRB and 20 (90.9%) were rated by the organization as slight/no therapeutic improvements and 2 (9.1%) as moderate therapeutic improvements. Health Canada issued safety advisories for 16 (25.8%) out of the 62 unique drugs in the 70 drug-indications, but only 5 drugs that had safety advisories also had PMPRB therapeutic evaluations, all of those being slight/no therapeutic improvement. Therefore, a separate Canadian assessment of the additional therapeutic value and safety was only possible for a small minority of not acceptable drugs, i.e., 5 of the 70 drug-indications were slight to no improvements and had serious safety issues.

### Relationship in time between EMA and Health Canada evaluations of not acceptable drugs

Twenty-seven of the 70 drug-indications were approved by Health Canada before the EMA/ANSM approval and 43 after the EMA/ANSM approval. Forty-six drug-indications were approved by Health Canada within one year before or after EMA/ANSM approval. The median difference, in days, between approval times was 129 (interquartile range -102, 341) in favour of the EMA/ANSM.

**Table 1. Drug-indications rated as "not acceptable" by Prescrire.**

| Generic name | Indication |
| --- | --- |
| Adalimumab | Moderate to severe hidradenitis suppurativa in adults |
| Alemtuzumab | Relapsing-remitting multiple sclerosis with active disease defined by clinical or imaging features |
| Ataluren | Duchene muscular dystrophy in children aged 2–4 |
| Ataluren | Duchene muscular dystrophy in children aged 5 and older |
| Atezolizumab | Inoperable or metastatic triple-negative breast cancer |
| Atezolizumab | Metastatic lung cancer |
| Bevacizumab | First-line treatment of certain types of lung cancer |
| Bevacizumab | In combination with carboplatin and paclitaxel for epithelial ovarian, fallopian tube or primary peritoneal cancer |
| Bevacizumab | In combination with carboplatin and paclitaxel with first recurrence of platinum-sensitive epithelial ovarian, fallopian tube or primary peritoneal cancer who have not received prior therapy with bevacizumab |
| Bezlotoxumab | Clostridium difficile |
| Blinatumomab | Adults with acute lymphoblastic leutemia in remission with residual leukemia cells |
| Brentuximab vedotin | Hodgkin lymphoma, after autologous transplantation |
| Cabozantinib | Medullary thyroid cancer |
| Canagliflozin | Monotherapy when diet and exercise do not provide adequate control in patients for whom metformin is not appropriate; add-on therapy with other glucose lowering products when these together do not provide adequate control |
| Canakinumab | Symptomatic treatment of adults with frequent gouty attacks in whom NSAIDs and colchicine contraindicated and corticosteroids not appropriate |
| Capsaicin | Peripheral neuropathic pain in adults with diabetes |
| Ceritinib | Non-small cell lung cancer |
| Ciclosporin eye drops | Treatment of severe keratitis in adults with dry eye disease |
| Cladribine | Multiple sclerosis |
| Colchicine + powdered opium + tiemonium | Acute pericarditis |
| Dabrafenib + Trametinib | Treatment of adults with advanced non-small cell lung cancer with a BRAF V600 mutation |
| Daclizumab | Multiple sclerosis |
| Dapagliflozin | Adjunct to insulin in patients with BMI > 27 when insulin alone does not provide adequate glycemic control |
| Dapagliflozin | Monotherapy where metformin inappropriate; add-on combination therapy with other products |
| Defibrotide | Treatment of severe hepatic veno-occlusive disease in hematopoetic stem-cell transplantation |
| Denosumab | Bone loss associated with long term systemic glucocorticoid therapy in adults with increased risk of fracture |
| Denosumab | Osteoporosis in men with increased risk of fracture |
| Drospirenone | Oral contraception |
| Eltrombopag | Thrombocytopenia in patients with hepatitis C |
| Equine estrogens + bazedoxifene | Estrogen deficiency symptoms in postmenopausal women with a uterus for who treatment with progestin not appropriate |
| Ertugliflozin (and combination products) | Type II diabetes |
| Esketamine | Treatment resistant depression |
| Esketamine | Depression with a high risk of suicide |
| Estetrol + drospirenone | Oral contraception |

*(Continued)*

**Table 1.** (Continued)

| Generic name | Indication |
|---|---|
| Febuxostat | Tumour lysis syndrome |
| Fenfluramine | Dravet syndrome |
| Ferumoxytol | Iron deficiency anemia in patients with chronic renal failure |
| Fusidic acid + betamethasone valerate | Eczematour skin rash when secondary bacterial infection is confirmed or suspected |
| Guanfacine | ADHD for whom stimulants are not suitable, not tolerated or ineffective |
| Idebenone | Treatment of visual impairment in adolescent and adult patients with Leber's Hereditary Optic Neuropahty |
| Inhaled mannitol | Treatment of cystic fibrosis in adults as add-on therapy |
| Insulin degludec + liraglutide | Treatment of Type 2 diabetes in combination with oral glucose-lower agents |
| Ivabradine + carvedilol | Heart failure |
| Linagliptin | Monotherapy for Type 2 diabetes; in combination with metformin when diet and exercise plus metformin do not provide adequate glycemic control |
| Liraglutide | Adolescents 12 and older with obesity and a body weight greater than 60 kg |
| Loxapine | Mild to moderate agitation in adults with schizophrenia or bipolar disorder |
| Mepolizumab | Add-on treatment for severe refractory eosinophilic asthma in adults |
| Mepolizumab | Add-on treatment for severe refractory eosinophilic asthma in adolescents and children 6 years and older |
| Misoprostol vaginal insert | Inducing labour |
| Naltrexone + bupropion | Adjunct to reduced calorie diet and increased physical activity for management of weight if BMI > 30 or 27 with weight-related comorbities |
| Nintedanib | Idiopathic pulmonary fibrosis |
| Nintedanib | Systemic sclerosis-associated interstitial lung disease |
| Nintedanib | Fibrosing interstitial lung disease in which cause of progresson unknown |
| Nivolumab | Hodgkin lymphoma |
| Obeticholic acid | Primary biliary cholangitis |
| Olaparib | Monotherapy in ovarian cancer without a BRCA mutation |
| Olaparib | Monotherapy for maintenance of patients who are in complete or partial response to platinum-based chemotherapy—ovarian cancer |
| Olaparib | Replacement for chemotherapy in certain metastatic pancreatic cancers |
| Olmesartan | Hypertension in children 6 and over |
| Omalizumab | Add-on for treatment of chronic spontaneous urticaria in adults and adolescents with inadequate response to H1 antihistamines |
| Omalizumab | Nasal polyposis |
| Ospemifene | Moderate to severe symptomatic vulvar and vaginal atrophy in post-menopausal women who are not candidates for local estrogen therapy |
| Ozanimod | Relapsing-remitting multiple sclerosis |
| Palbociclib | In combination with aromatase inhibitor or estrogen antagonist fulvestrant in advanced or metastatic breast cancer |
| Panobinostat | Multiple myeloma |
| Pazopanib | Treatment of selective subtypes of advanced soft-tissue sarcoma who have received prior chemotherapy or who have progressed |
| Peanut protein | Oral desensitization |

(*Continued*)

**Table 1.** (Continued)

| Generic name | Indication |
|---|---|
| Pegloticase | Treatment of severe debilitating gout iin adults who may also have erosive joint involvement and who have failed to normalize serum uric acid |
| Pembrolizumab | Hodgkin lymphoma |
| Pemigatinib | Monotherapy for locally advanced or metastatic cholangiocarcinoma that has progressed after at least one prior line of systematic therapy |
| Pentosan polysulfate | Bladder pain syndrome |
| Pertuzumab | Combined with trastuzumab and chemotherapy in treatment of breast cancer at high risk of recurrence |
| Pertuzumab | In combination with trastuzumab and chemotherapy for neoadjuvant treatment |
| Pirfenidone | Mild to moderate ioiopathic pulmonary fibrosis |
| Polatuzumab vedotin | Large B-cell lymphoma |
| Ponesimod | Relapsing-remitting multiple sclerosis |
| Prucalopride | Chronic constipation in men |
| Recombinant human parathyroid hormone | Chronic hypoparathyroidism not adequately controlled by standard therpay |
| Regorafenib | Gastrointestinal stromal tumours after treatment failure |
| Ribociclib | Locally advanced breast or metastatic breast cancer |
| Rivaroxaban | Prevention of atherothrombotic events in adults after an acute coronary syndrome |
| Roflumilast | Maintenance treatment of severe chronic obstructive disease associated with chronic bronchitis |
| Romosozumab | Severe postmenopausal osteoporosis |
| Saxagliptin | Monotherapy when metformin is inappropriate; triple therapy in combination with metformin plus a sulphonylurea or insulin |
| Saxagliptin + metformin | Type II diabetes inadequately controlled on metformin alone or those already treated with saxagliptin and metformin separately |
| Selexipag | Pulmonary arterial hypertension |
| Simvastatin + fenofibrate | Adults with high cardiovascular risk with mixed dyslipidemia |
| Siponimod | Secondary progressive multiple sclerosis |
| Solriamfetol | Excessive daytime sleepiness in adults with narcolepsy or obstructive sleep apnea |
| Sorafenib | Treatment of progressive, locally advanced or metastatic thyroid cancinoma refractory to radioactive iodine |
| Strontium ranelate | Treatment of osteoporosis in adult men at increased risk of fracture |
| Telavancin | Treatment of nosocomial pneumonia caused by methicillin-resistant S. aureus |
| Tenofovir aalfenamide + emtricitabine + elvitegravir + cobicistat | HIV in children |
| Teriflunomide | Relapsing-remitting multiple sclerosis |
| Tigecycline | Complicated skin and soft tissue infections in children and complicated intra-abdominal infections |
| Tolvaptan | Slow progression of cyst development and renal insufficiency of autosomal dominant polycystic kidney disease |
| Trastuzumab (subcutaneous) | Metastatic and non-metastatic breast cancer overexpressing HER-2 protein |
| Varenicline | Explicitly encouraged additional course of treatment after failure to quit smoking during initial therapy |
| Vildagliptin | Monotherapy in Type II diabetes for whom metformin is inappropriate |

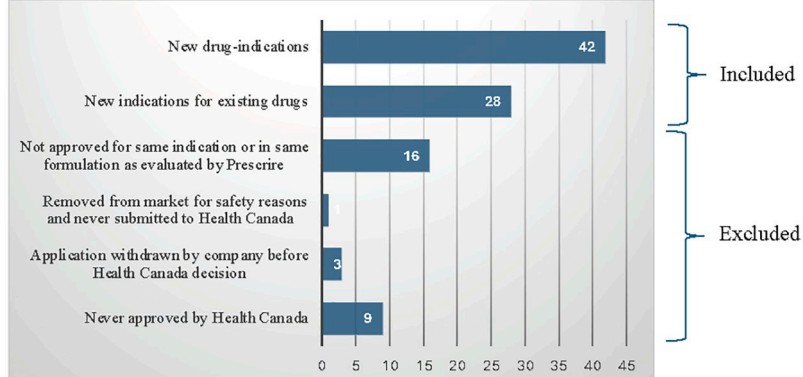

**Fig 1. Status of "not acceptable" drugs in Canada.** Source: Author's calculations based on information from Prescrire International and Health Canada databases.

## Discussion

Seventy (70.7%) of the 99 drug-indications approved by either the EMA or ANSM and rated by Prescrire as not acceptable were approved by Health Canada. Of the 42 new drug-indications, 22 had therapeutic ratings and 20 (90.9%) were evaluated as offering slight/no therapeutic improvement. The PMPRB therapeutic evaluations available indicate that the large majority of not acceptable new drug-indications approved by Health Canada are only slight or no improvements over existing drugs as rated by the PMPRB and there were safety advisories for a quarter of the unique drugs, but the number of people possibly negatively affected by these drugs cannot be determined from the safety advisories. So, while not acceptable drug-indications pose a risk to patients, the degree of risk cannot yet be quantified.

The presence of a majority of drugs rejected by Prescrire is largely congruent with a previous analysis that found that out of Prescrire's list of 92 drugs that should be avoided, 56 (60.9%) were marketed in Canada in the same formulation and for the same indication as the products that Prescrire evaluated. Thirty-six of those 92 drugs were evaluated by the PMPRB; only two were classed as breakthrough or substantial improvements and 3 were moderate improvements, the remaining 31 were slight/no improvements [6].

The results from this study point to the diversity of conclusions about the benefit to harm ratio between those given by regulatory agencies like EMA/ANSM and Health Canada and those given by Prescrire. The difference is unlikely to be due to the evidence available to the organizations. Prescrire did its evaluations at about the same time as the EMA/ANSM approvals and 46 of Health Canada's approvals were within 1 year of the approval by EMA/ANSM. However, Prescrire compares the benefit to harm ratio of new drug-indications with that of existing products whereas drug regulatory authorities look at the evidence for a new drug-indication in isolation. Therefore, the Prescrire rejection of certain drug-indications may not be applicable in Canada because of differences in the availability of alternative treatments in France and Canada.

There have also been concerns about a possible decline in Health Canada's regulatory standards that may contribute to the approval of drug-indications that have a less favourable benefit to harm ratio. A 2023 study looking at Health Canada standards over a 10-year period found a decrease in the number of pivotal trials necessary to get a new drug approved along with a decrease in the percent of Phase 3 clinical trials and a decrease in the percent of trials that were randomized, controlled, and blinded [15]. It is difficult to disentangle whether these

changes reflect less rigorous standards, an adaptation to the larger number of orphan drugs being submitted, the increasing presence of novel products such as those based on advanced cell and gene therapy that require the use of customized regulatory requirements or a combination of all these reasons.

## Limitations

The percent of not acceptable drug-indications available only applies to the Canadian context and may be different in other jurisdictions. Although Prescrire's ratings of drug-indications are generally similar to those produced by other organizations, there is no independent confirmation specifically of its rating of "not acceptable" drugs. The assessment of the benefit to harm ratio of drugs occurs early in a drug's lifecycle and may change over time but Prescrire only occasionally does second evaluations of drug-indications. All the data were gathered by a single individual and that may have inadvertently introduced errors in the transcription of data.

## Conclusion

A majority of drugs considered not acceptable by Prescrire have been approved by Health Canada and there were safety advisories for a quarter of them. To what degree those drugs leads to more harms than benefits for patients who are taking them needs to be urgently investigated. Finally, the reasoning behind Health Canada's approval of these drugs should be interrogated.

## Supporting information

**S1 Table. Complete data.**
(XLSX)

## Author Contributions

**Conceptualization:** Joel Lexchin.

**Data curation:** Joel Lexchin.

**Formal analysis:** Joel Lexchin.

**Methodology:** Joel Lexchin.

**Writing – original draft:** Joel Lexchin.

**Writing – review & editing:** Joel Lexchin.

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
