## [Decision Letter · Decision Letter 0]

30 May 2024

PONE-D-24-11119Status of “not acceptable” drug-indications in Canada: a cohort studyPLOS ONE

Dear Dr. Lexchin,

Thank you for submitting your manuscript to PLOS ONE. After careful consideration, we feel that it has merit but does not fully meet PLOS ONE’s publication criteria as it currently stands. Therefore, we invite you to submit a revised version of the manuscript that addresses the points raised during the review process.

**Please see the reviewer comments (including, for one reviewer, in the attachment) below. **

We look forward to receiving your revised manuscript.

Kind regards,

Dzintars Gotham

Academic Editor

PLOS ONE

Journal Requirements:

Between 2020-2024, Joel Lexchin received payments for writing a brief on the role of promotion in generating prescriptions for a legal firm, for being on a panel about pharmacare and for co-writing an article for a peer-reviewed medical journal. He is a member of the Board of Canadian Doctors for Medicare. He receives royalties from University of Toronto Press and James Lorimer & Co. Ltd. for books he has written. He is participating in research funded by the Canadian Institutes of Health Research.

We note that one or more of the authors are employed by a commercial company. 

“The funder provided support in the form of salaries for authors, but did not have any additional role in the study design, data collection and analysis, decision to publish, or preparation of the manuscript. The specific roles of these authors are articulated in the ‘author contributions’ section.”

Reviewers' comments:

Reviewer's Responses to Questions

**Comments to the Author**

1. Is the manuscript technically sound, and do the data support the conclusions?

Reviewer #1: Yes

Reviewer #2: No

2. Has the statistical analysis been performed appropriately and rigorously? 

Reviewer #1: N/A

Reviewer #2: Yes

3. Have the authors made all data underlying the findings in their manuscript fully available?

Reviewer #1: Yes

Reviewer #2: Yes

4. Is the manuscript presented in an intelligible fashion and written in standard English?

Reviewer #1: Yes

Reviewer #2: No

5. Review Comments to the Author

**Reviewer #1:** I very much enjoyed the paper.

It is a simple - but far from simplistic - comparison of drug approvals, using Prescrire as the accepted external standard. It answers a worthy regulatory question - what is being approved and if indications are sound - even if the 'why' cannot be ascertained. This work has the potential for application in multiple contexts.

It also makes an important case against the sometimes perilous application of reliance - perhaps not the intention of the paper, but certainly enhances the point that independent analyses are something regulators should not discard.

Two issues to consider:

1) The meaning of the term 'minor value' (Discussion Section, page 15, line 5) is not clear. Please explain or maybe amend;

2) Please consider adding the material in the Supplement to the core text. The table makes all the difference and is a crucial element for readers to fully understand the depth of this analysis. Without it numbers and proportions do not really get to the point. Readers must see the drugs and indications.

In closing, Prescrire´s much more holistic outlook - in contrast to the isolated view of RAs - is what we aim for - and consistent with appropriateness of drug use and with the EM concept, which drug approval processes worldwide must go back to, if only for the sake of equity and sustainability, if not for upholding adequate standards of care.

**Reviewer #2:** Dear Author, dear Editors,

This study aims to assess the status of drugs in Canada that have been rated as ‘not acceptable’ by the French drug bulletin Prescrire. In general, the manuscript adheres to most of PLOS ONE’s publication criteria. However, especially the analyses performed and the reporting and contextualization of the results require some more depth. Therefore, I deem several major revisions necessary before I can recommend it for publication. In that light, I have several questions and suggestions for the author to clarify the study and improve the analyses and the presentation of the results.

Please see the attached document for the specific comments.

6. PLOS authors have the option to publish the peer review history of their article (what does this mean?). If published, this will include your full peer review and any attached files.

Reviewer #1: **Yes: **Claudia Garcia Serpa Osorio-de-Castro

Reviewer #2: **Yes: **Lourens Bloem, PharmD, PhD

---

## [Author Response · Author response to Decision Letter 0]

31 May 2024

Response to editors

Please ensure that your manuscript meets PLOS ONE's style requirements, including those for file naming. The PLOS ONE style templates can be found at  https://journals.plos.org/plosone/s/file?id=wjVg/PLOSOne_formatting_sample_main_body.pdf and  https://journals.plos.org/plosone/s/file?id=ba62/PLOSOne_formatting_sample_title_authors_affiliations.pdf

I have carefully read PLOS ONE’s style requirements and can confirm that my manuscript meets those requirements.

2. Thank you for stating the following in the Competing Interests section:  Between 2020-2024, Joel Lexchin received payments for writing a brief on the role of promotion in generating prescriptions for a legal firm, for being on a panel about pharmacare and for co-writing an article for a peer-reviewed medical journal. He is a member of the Board of Canadian Doctors for Medicare. He receives royalties from University of Toronto Press and James Lorimer & Co. Ltd. for books he has written. He is participating in research funded by the Canadian Institutes of Health Research.   We note that one or more of the authors are employed by a commercial company. 

Please read my COI statement carefully. It does NOT say that I am or have been employed by a commercial entity. Receiving payment for being on a panel or for writing a brief does NOT constitute employment. Both of the times that I received a payment it was based on a contractual arrangement – I provided a service and received payment.

See my above response. Once again, I was not and am not employed by any commercial entity. The study was NOT funded. The absence of any funding was already stated when I initially submitted the manuscript.

Please also include the following statement within your amended Funding Statement.  “The funder provided support in the form of salaries for authors, but did not have any additional role in the study design, data collection and analysis, decision to publish, or preparation of the manuscript. The specific roles of these authors are articulated in the ‘author contributions’ section.” If your commercial affiliation did play a role in your study, please state and explain this role within your updated Funding Statement.

Please seem my above response. The study was NOT funded. The absence of any funding was already stated when I initially submitted the manuscript. 

Please see my above response. There is no other commercial affiliation to declare beyond what is already contained in the COI statement.

Please carefully read what I have written above. The study was NOT funded. I was not and am NOT 

employed by any commercial entity.

The reference list has been reviewed and it is complete and correct.

Response to Reviewer 1

I very much enjoyed the paper. It is a simple - but far from simplistic - comparison of drug approvals, using Prescrire as the accepted external standard. It answers a worthy regulatory question - what is being approved and if indications are sound - even if the 'why' cannot be ascertained. This work has the potential for application in multiple contexts. It also makes an important case against the sometimes perilous application of reliance - perhaps not the intention of the paper, but certainly enhances the point that independent analyses are something regulators should not discard. Two issues to consider: 

I appreciate the support from the reviewer for this article.1) The meaning of the term 'minor value' (Discussion Section, page 15, line 5) is not clear. Please explain or maybe amend; 

“Minor value” has been replaced by the phrase “only slight or no improvements over existing drugs as rated by the PMPRB”.

2) Please consider adding the material in the Supplement to the core text. The table makes all the difference and is a crucial element for readers to fully understand the depth of this analysis. Without it numbers and proportions do not really get to the point. Readers must see the drugs and indications. 

The Supplementary Table is very large. If the editors request that it be included as part of the main article I would be happy to do so, but in the meantime I have added a table that gives the generic name and indications of the 99 drug-indications rated by Prescrire as not acceptable. 

In closing, Prescrire´s much more holistic outlook - in contrast to the isolated view of RAs - is what we aim for - and consistent with appropriateness of drug use and with the EM concept, which drug approval processes worldwide must go back to, if only for the sake of equity and sustainability, if not for upholding adequate standards of care.

Response to Reviewer 2

This study aims to assess the status of drugs in Canada that have been rated as ‘not acceptable’ by the French drug bulletin Prescrire. In general, the manuscript adheres to most of PLOS ONE’s publication criteria. However, especially the analyses performed and the reporting and contextualization of the results require some more depth. Therefore, I deem several major revisions necessary before I can recommend it for publication. In that light, I have several questions and suggestions for the author to clarify the study and improve the analyses and the presentation of the results.

Major

• Although highly specialised readers in this field may instantly appreciate the choice of using Prescrire’s ratings of drugs, the manuscript lacks a clear rationale why this study aims to use data from Prescrire for the Canadian context. The introduction scarcely mentions what Prescrire is. 

I have added a brief description of Prescrire along with a reference in the first paragraph in the Introduction as well as an explanation of why Prescrire ratings are being used: “Although bodies such as the French Haute Autorité de Santé and the Institute for Quality and Efficiency in Health Care, the German health technology assessment agency, evaluate the therapeutic value of new drugs using an ordinal scale and publish their findings in English, the evaluations from Prescrire are used in this study because it is the only organization that names drugs that should not have been given regulatory approval.”

• Similarly, the Limitations section (p.11) states that there is no independent confirmation of Prescrire’s ratings. However, several studies have compared therapeutic value as assessed by Prescrire with that assessed by (European) HTA agencies. Their findings add important context and shed light on the generalisability of Prescrire’s ratings. I thus recommend the author to clarify the choice of using Prescrire to make the study more accessible and understandable for non-specialists as well as to readers who are familiar with assessing the therapeutic value of new drugs.

Prescrire’s ratings have been compared to those from the bulletin of the Swedish Medical Products Agency and from the French regulatory agency Haute Autorité de Santé and I have cited this literature. I have also modified the statement in the Limitations section.

• While a lot of studies have evaluated therapeutic value of new drugs as assessed by (different) regulatory authorities and differences between them and HTA organisations, the manuscript lacks grounding in and connection of its findings and rationale to these studies and discussions. I believe this would add to the manuscript’s relevance and rationale when these discussions are referred to and reflected upon both in the introduction and the discussion section.

Prescrire’s ratings are compared to those produced by the Patented Medicine Prices Review Board (PMPRB), but the PMPRB is not a health technology assessment body. (That function is assumed by the Canadian Agency for Drugs and Technologies in Health.) The PMPRB assesses the additional therapeutic value of new patented drugs compared to drugs already on the Canadian market and that assessment is used in the process of setting a maximum introductory price for new patented drugs. Therefore, I have not discussed the relationship between the regulatory approvals made by Health Canada and HTA recommendations, but I have made the role of the PMPRB clear. 

• The study’s Methods are in principle fit for answering the presented research question, yet some explanation would help to understand why and which steps were undertaken. For instance, it is yet unclear why data about the completion of the post-authorisation conditions were collected for drugs with a NOC/c approval and why the safety advisories were collected. Moreover, providing more context and detail to the data analysis section would help in understanding why steps are taken.

Information about the type of approval given by Health Canada has now been removed from the Methods and Results sections. 

The Methods section contains an explanation about why information was collected about safety reports and the PMPRB therapeutic evaluations: “An attempt was made to produce a Canadian assessment of the benefit to harm ratio of not acceptable drugs on the Canadian market to evaluate how these assessments compared to the ones from Prescrire. Accordingly, the Recall and Safety Alerts website was searched for safety advisories issued by Health Canada for drugs in this group [11] accompanied by a search of therapeutic evaluations done by the Patented Medicine Prices Review Board (PMPRB) and published in its annual reports [12]. The PMPRB is an independent quasi-judicial body that reviews the prices of new patented medicines sold in Canada and as part of that function undertakes therapeutic evaluations and categorizes the drugs as breakthrough, substantial improvement, moderate improvement, slight or no improvement. (The PMPRB does not evaluate the therapeutic value of new indications for existing drugs.)”

The Methods section also now explains why the difference in approval dates was calculated between the EMA and Health Canada: “Prescrire evaluations and Health Canada approvals may have differed due to the two organizations being presented with different information. Prescrire evaluations are undertaken shortly after EMA approval and therefore, the difference between the EMA and Health Canada approval dates (in days) was calculated and the median and interquartile range was reported to explore this possibility.”

• In relation, the completion status and safety advisories not necessarily only lead to drug withdrawals, although the last sentence of the first Results paragraph (p.8, first sentence) and the second-last Results paragraph do suggest this. This should be further explained.

The last sentence in the first paragraph in the Results section was not intended to mean that drugs with safety issues are always withdrawn from the market. The sentence was part of the description about the EMA regulatory status of the not acceptable drugs. The second last Results paragraph has been deleted.

• In general, the Results section requires restructuring and additional headings to better guide the reader through the study’s findings and their relevance.

Additional subheadings have been added in the Results section. The subsection in the Results “Not acceptable drugs approved by Health Canada” has been restructured to make it easier to follow.

• To support and make the study’s findings more intelligible, the Results section would benefit from adding a table and/or figure (such as bar or pie charts) in which the findings are presented. Similarly, a typical ‘Table 1’ would provide required information on the drugs and indications included in the study (n=62/70). Figure 1 would benefit from a clearer visual distinction between the included and excluded drugs. Of note, the ‘safety advisories issues’ now suggest exclusion of these drugs from the cohort. Moreover, clarifying how the results and percentages relate to the different ‘buckets’ of medicines would be helpful for readers.

Table 1 giving the generic names and indications of the not acceptable drugs approved by the EMA and the French regulatory authority has been added. Figure 1 has been replaced with a new bar chart that also clearly indicates the reasons why drugs were included and excluded.

Minor

• I recommend to revise the title and abstract to clarify that the ‘not acceptable’ rating is as considered by Prescrire. Now, they suggest that the study assessed drugs that were refused for marketing authorization by Health Canada. This should also be clarified throughout the manuscript.

The title has been changed and now reads: “Were drug-indications evaluated by Prescrire as “not acceptable” approved by Health Canada: a cohort study”. The manuscript makes it clear that a minority of these drugs were refused market authorization by Health Canada.

• As mentioned in the Limitations section, the data collection has been performed only by the author himself. Given the vast number of bulletins that needed to be checked manually, I would like to emphasise the importance to validate at least a sample by a second researcher.

I appreciate the concern of the reviewer. In the absence of a second researcher, I have gone through the issues of Prescrire again to verify that all of the drugs rated as not acceptable have been included.

• I must admit that I could not locate the difference in appraisals between Prescrire and Health Canada as discussed in the second sentence of the abstract, in the Results section. At least, it was not explicitly discussed.

Health Canada does not offer an appraisal of the therapeutic value of new drugs, it either approves them for marketing or rejects them. (At times, the manufacturer withdraws the submission before Health Canada has reached a decision.) The Canadian status of the 99 drug-indications is described in the Results subsection “Not acceptable drugs approved by Health Canada”.

• The last paragraph of the Results (“Twenty-seven of the (…) in favour of the EMA/ANSM” (p.9) may be moved more forward, for a more logical flow o

---

## [Decision Letter · Decision Letter 1]

21 Jun 2024

PONE-D-24-11119R1Were drug-indications evaluated by Prescrire as "not acceptable" approved by Health Canada: a cohort studyPLOS ONE

Dear Dr. Lexchin,

Thank you for submitting your manuscript to PLOS ONE. After careful consideration, we feel that it has merit but does not fully meet PLOS ONE’s publication criteria as it currently stands. Therefore, we invite you to submit a revised version of the manuscript that addresses the points raised during the review process. Please see the reviewers' comments.

We look forward to receiving your revised manuscript.

Kind regards,

Dzintars Gotham

Academic Editor

PLOS ONE

Journal Requirements:

Reviewers' comments:

Reviewer's Responses to Questions

**Comments to the Author**

1. If the authors have adequately addressed your comments raised in a previous round of review and you feel that this manuscript is now acceptable for publication, you may indicate that here to bypass the “Comments to the Author” section, enter your conflict of interest statement in the “Confidential to Editor” section, and submit your "Accept" recommendation.

Reviewer #1: All comments have been addressed

Reviewer #2: (No Response)

2. Is the manuscript technically sound, and do the data support the conclusions?

Reviewer #1: Yes

Reviewer #2: Partly

3. Has the statistical analysis been performed appropriately and rigorously? 

Reviewer #1: Yes

Reviewer #2: N/A

4. Have the authors made all data underlying the findings in their manuscript fully available?

Reviewer #1: Yes

Reviewer #2: Yes

5. Is the manuscript presented in an intelligible fashion and written in standard English?

Reviewer #1: Yes

Reviewer #2: No

6. Review Comments to the Author

Reviewer #1: Table 1 was a great gain to the paper.

Clarification of the term 'minor value' was resolved with substitution.

I did not encounter the difficulties presented by Reviewer 2, but no harm was done with the answers to the multiple queries.

The manuscript is sound and well-presented.

Reviewer #2: Dear Author, dear Editors,

I thank the author for the response to the comments I made on the previous version of the manuscript. Several of the issues have been adequately addressed and resolved. However, based on the responses, I deem several revisions still necessary before I can recommend it for publication. I have several outstanding questions and suggestions for the author to clarify the study and improve the analyses and the presentation of the results. Please refer to the attachment.

Kind regards

7. PLOS authors have the option to publish the peer review history of their article (what does this mean?). If published, this will include your full peer review and any attached files.

Reviewer #1: **Yes: **Claudia Garcia Serpa Osorio-de-Castro

Reviewer #2: **Yes: **Lourens T. Bloem

---

## [Author Response · Author response to Decision Letter 1]

22 Jun 2024

I am grateful to the reviewers for the time that they have taken to examine the manuscript.

Reviewer #1

Table 1 was a great gain to the paper.

Clarification of the term 'minor value' was resolved with substitution.

I did not encounter the difficulties presented by Reviewer 2, but no harm was done with the answers to the multiple queries.

The manuscript is sound and well-presented.

I appreciate the reviewer’s confidence in the quality of the article.

Reviewer #2

Major

• I strongly recommend to better ground the manuscript in and connect its rationale and findings to the larger discussion in the scientific literature about the discrepancies between regulatory approvals (e.g., by the EMA or Health Canada) and the added value for society of new drugs. Prescrire is not the only organisation that assesses the therapeutic/clinical/added(/societal?) value of new drugs, but a discussion of, for example, discrepancies between regulatory and HTA evaluations or the role of other value scales is lacking. This is crucial to understand the broader context of the manuscript and to strengthen its rationale.

I appreciate the reviewer’s point about understanding the overall value of drugs to society, but that is not the goal of this manuscript. The primary goal is to examine whether Health Canada has approved drugs that Prescrire has determined should not have been approved by drug regulatory agencies. Importantly, drug regulatory agencies and health technology agencies (HTAs) serve two distinct functions. Drug regulatory agencies evaluate the results of pivotal trials to decide on the efficacy of the new product (i.e., is the drug superior to placebo) and is it safe enough to be used for its intended indication. HTAs evaluate whether a drug delivers value for the price and depending on the indication the cost per QALY may be variable. Therefore, a drug that the regulatory agency approves may be rejected for funding by agency performing the HTA because of low efficacy (but still superior to a placebo), high price or a combination of both. I would like the academic editor to take a position about whether I should follow through with the request from the reviewer.

• An explanation is required in the introduction and/or the methods section to understand why the author chose to use the PMPRB ratings and not those of the Canadian Agency for Drugs and Technologies in Health (CADTH), as well as how their position and evaluations relate to each other.

The PMPRB evaluates the additional therapeutic value of new drugs. CADTH makes recommendations about whether new drugs for particular indications should be funded. These are two different types of activities. CADTH might recommend reimbursement for a drug that is no better or worse than existing products as long as its price is reasonable or recommend not funding a drug not because the drug has a negative benefit to harm ratio but because the drug does not represent value for money. CADTH’s recommendations about funding cannot be directly translated into a categorical rating of the additional therapeutic value of drugs. Since the manuscript does not mention CADTH I do not feel the need to discuss its role in the context of drug funding in Canada.

• The author now clarified that he aimed to simulate a ‘Canadian benefit to harm ratio’ by combining PMPRB ratings and whether safety advisories were issued. This was not clear from the initial manuscript. However, it begs the question what the safety advisories add to this assessment because:

1. Doesn’t PMPRB already consider safety aspects in their evaluation of therapeutic improvement? If they do, please explain the added value of the safety advisories to the consideration of safety by the PMPRB.

The PMPRB considers the efficacy and safety of new drugs largely based on the documentation submitted by companies at the time when they apply for approval. Safety advisories are typically based on new information about issues that arise post-marketing and would not have been identified in the material submitted for approval.

2. My understanding of the safety advisories is that they are distributed post-approval by Health Canada. However, the drug-indication combinations in the cohort did not have similar follow-up and thus not the same risk time to receive a safety advisory. If correct, this could substantially bias the analysis. 

The reviewer is correct that safety advisories are issued post-approval based on newly identified risks associated with the drug. As stated in the Methods section, a search was made for any post-market safety advisories issued for all the drug-indications approved by Health Canada. Therefore, all the drug-indications approved by Health Canada had the same follow-up in terms of looking for new safety issues. 

3. In addition, the author specifically highlights for the drug-indication combinations for which safety advisories were issued, the proportion for which a PMPRB evaluation was available. This also biases the analysis and should be the other way around: n safety advisories/n PMPRB evaluations. 

I am unclear why the reviewer thinks that what I have said about the number of drugs with PMPRB evaluations and the number with safety evaluations biases the analysis. I believe that the revised final sentence on page 13 is clear: “Therefore, a separate Canadian assessment of the therapeutic value and safety was only possible for a small minority of not acceptable drugs, i.e., 5 of the 70 drug-indications were slight to no improvements and had serious safety issues.” 

Please correct me if the above is incorrect and explain why. If not, please revise the manuscript to address these issues.

• Page 13 states that “1 drug had been removed from the market for safety reasons”. The manuscript currently suggests that this drug was excluded from the analysis, while, if ever approved by Health Canada, it should have been included and carried forward for analysis in the section about “Canadian evaluations”. Please revise the manuscript and Figure 1 accordingly.

Presumably the reviewer is referring to fenfluramine. The text and Figure 1 have been revised to state that the drug was removed from the market and never submitted to Health Canada for approval.

• I would like to urge the author to adapt the newly added and very lengthy Table to a more general ‘Table 1’ with some aggregated characteristics to describe the cohort. For instance by including the following elements from the supplementary material: disease areas, period of approval and whether it was initially approved by EMA and/or ASNM, ND/NI.

Adapting the table in the manner requested by the reviewer would create a very large table. Making this change could easily be done, but I would appreciate guidance from the academic editor regarding whether he agrees with this request.

• Additionally, a flowchart that visualises which subset of drug-indications are used for the several analyses throughout the manuscript would substantially increase the intelligibility of the manuscript.

Figure 1 provides information and a count about the number of drugs approved by Health Canada and the reasons why drug-indications were excluded. There are only two other analyses undertaken. One is a count of the number of drugs with PMPRB evaluations and safety warnings and the second is the difference in approval times between Health Canada and the EMA. I don’t believe that a flowchart is necessary to be able to follow these analyses, but once again would appreciate guidance from the academic editor.

Minor

• The study’s objective is not solely about the approval by Health Canada but the title does suggest this. Please change the title to bring it more in line with the study’s objective and broader scope, for instance to read: “Status of not acceptable drugs as considered by Prescrire in Canada: a cohort study”.

The title has been changed to “Canadian status of “not acceptable” drugs as evaluated by Prescrire: a cohort study”.

• The manuscript’s intelligibility would profoundly benefit from more detail and explanation throughout the manuscript. Please consider the following points:

o Mention in the introduction that previous research has shown that drugs that should be avoided according to Prescrire are in fact available in Canada. This adds to the rationale and context for the study too. (about the reference Lexchin et al., 2017)

The change suggested by the reviewer has been made near the end of the Introduction.

o Introduction: the wording of ‘by the EMA’s decentralised procedure … a single market’ is not correct here, please adapt: “approved for marketing in France either by ASNM or the EMA’.

The end of the paragraph now reads “The date of approval of the drug-indications marketed in France was found by searching on the drug’s name on either the EMA website [7] or the website of the French Agence nationale de séurité du medicament et des produits de santé (ANSM) [8].”

o The author performs no assessment of the ‘benefit to harm ratio’ (like the EMA or Health Canada assess the benefit-risk), but he performs rather a ‘Canadian assessment of the added/clinical/therapeutic value’. Please clarify this accordingly throughout the manuscript.

The reviewer is not correct. I do not attempt to perform “a Canadian assessment of the added/clinical/therapeutic value”. What I have done is to look at additional therapeutic value as assessed by the PMPRB and safety based on safety advisories. However, I have changed the opening sentence of the paragraph on page 6 to read “An assessment was made of the additional therapeutic value and the safety of not acceptable drugs on the Canadian market.” A similar change was made in the “Canadian evaluations of the therapeutic benefits and harms of not acceptable drug-indications” subsection of the Results.

o The Data Analysis section should be revised to clarify which analyses were performed and how.

There are only three analyses that are done. The first is a count of the number of not acceptable drugs that Health Canada has approved. The second are counts of the number of drugs that have been evaluated by the PMPRB and that have safety warnings and the third is a calculation of the difference in the time between when the EMA and Health Canada approved the drugs. The first and third analyses are already described in the Data Analysis section. I have now added that counts were made of the number of drugs with PMPRB evaluations and with safety warnings. 

o In the Results section ‘Not acceptable drug approved by Health Canada’, adding that the 99 drug-indications concern those that were deemed ‘not acceptable’ by Prescrire would add clarity about the cohort characteristics.

The first sentence in this section now reads “Out of the 99 drug-indications identified by Prescrire as not acceptable…”

o The relevance for extracting and reporting the drug formulation is made clear in the author’s answer to the question. Please add this reasoning to the methods section, as well as the fact that the formulation was extracted for each drug, in addition to the generic name, brand name, etc.

The information about why the formulation of the drug was recorded has been added to the Methods section.

---

## [Editor Report · Decision Letter 2]

17 Jul 2024

Canadian status of "not acceptable" drugs as evaluated by Prescrire: a cohort study

PONE-D-24-11119R2

Dear Dr. Lexchin,

We’re pleased to inform you that your manuscript has been judged scientifically suitable for publication and will be formally accepted for publication once it meets all outstanding technical requirements.

Kind regards,

Dzintars Gotham

Academic Editor

PLOS ONE
---

## [Editor Report · Acceptance letter]

23 Jul 2024

PONE-D-24-11119R2 

PLOS ONE

Dear Dr. Lexchin, 

I'm pleased to inform you that your manuscript has been deemed suitable for publication in PLOS ONE. Congratulations! Your manuscript is now being handed over to our production team.

Kind regards, 

on behalf of

Dr. Dzintars Gotham 

Academic Editor

PLOS ONE